

# Exploring the trends of adaptation and evolution of sclerites with regards to habitat depth in sea pens

Yuka Kushida[1,2,3,4], Yukimitsu Imahara[2,5,6], Hin Boo Wee[7], Iria Fernandez-Silva[8], Jane Fromont[9], Oliver Gomez[9], Nerida Wilson[9,10], Taeko Kimura[11], Shinji Tsuchida[12], Yoshihiro Fujiwara[12], Takuo Higashiji[13], Hiroaki Nakano[14], Hisanori Kohtsuka[15], Akira Iguchi[2,16] and James Davis Reimer[4,17]

[1] Faculty of Geo-Environmental Science, Rissho University, Kumagaya, Saitama, Japan
[2] Geological Survey of Japan, National Institute of Advanced Industrial Science and Technology (AIST), Tsukuba, Ibaraki, Japan
[3] International Center for Island Studies Amami Station, Kagoshima University, Amami, Kagoshima, Japan
[4] Molecular Invertebrate Systematics and Ecology Laboratory, Graduate School of Engineering and Science, University of the Ryukyus, Nishihara, Okinawa, Japan
[5] Kuroshio Biological Research Foundation, Otsuchi, Kochi, Japan
[6] Octocoral Research Laboratory, Wakayama, Wakayama, Japan
[7] Institut Perubahan Iklim, Universiti Kebangsaan Malaysia, Selangor Darul Ehsan, Malaysia
[8] Department of Biochemistry, Genetics and Immunology, Campus Universitario, University of Vigo, Vigo, Spain
[9] Collections & Research, Western Australian Museum, Welshpool, Western Australia, Australia
[10] School of Biological Sciences, University of Western Australia, Perth, Western Australia, Australia
[11] Department of Life Sciences, Graduate School of Bioresources, Mie University, Tsu, Mie, Japan
[12] Japan Agency for Marine-Earth Science and Technology (JAMSTEC), Yokosuka, Kanagawa, Japan
[13] Okinawa Churaumi Aquarium, Okinawa Churashima Foundation, Motobu, Okinawa, Japan
[14] Shimoda Marine Research Center, University of Tsukuba, Shimoda, Shizuoka, Japan
[15] Misaki Marine Biological Station, Graduate School of Science, University of Tokyo, Miura, Kanagawa, Japan
[16] Research Laboratory on Environmentally-Conscious Developments and Technologies [E-Code], National Institute of Advanced Industrial Science and Technology (AIST), Tsukuba, Ibaraki, Japan
[17] Tropical Biosphere Research Center, University of the Ryukyus, Nishihara, Okinawa, Japan

Corresponding author
Yuka Kushida, ykushida@ris.ac.jp

## ABSTRACT

Octocorals possess sclerites, small elements comprised of calcium carbonate ($CaCO_3$) that are important diagnostic characters in octocoral taxonomy. Among octocorals, sea pens comprise a unique order (Pennatulacea) that live in a wide range of depths. Habitat depth is considered to be important in the diversification of octocoral species, but a lack of information on sea pens has limited studies on their adaptation and evolution across depth. Here, we aimed to reveal trends of adaptation and evolution of sclerite shapes in sea pens with regards to habitat depth *via* phylogenetic analyses and ancestral reconstruction analyses. Colony form of sea pens is suggested to have undergone convergent evolution and the loss of axis has occurred independently across the evolution of sea pens. Divergences of sea pen taxa and of sclerite forms are suggested to depend on habitat depths. In addition, their sclerite forms may be related to evolutionary history of the sclerite and the surrounding chemical environment as well as water temperature. Three-flanged sclerites may possess the tolerance towards the environment of the deep sea, while plate sclerites are suggested to be adapted towards shallower waters, and have evolved independently multiple times.

The common ancestor form of sea pens was predicted to be deep-sea and similar to family Pseudumbellulidae in form, possessing sclerites intermediate in form to those of alcyonaceans and modern sea pens such as spindles, rods with spines, and three-flanged sclerites with serrated edges sclerites, as well as having an axis and bilateral traits.

## INTRODUCTION

The order Pennatulacea, within the subclass Octocorallia (Cnidaria: Anthozoa), includes the sea pens and sea pansies. Sea pens are unique octocorals as they have adapted to a life on marine soft bottoms, and it is estimated there are about 200 species in the world's oceans (*Fabricius & Alderslade, 2001*; *Williams, 2011*). The primary polyp, called an oozooid, is the central part of the sea pen colony and includes the rachis and peduncle, and is a characteristic trait. The peduncle enables sessile life by acting as an anchor *via* embedding in soft sediments. The rachis hosts two to four different types of polyps, including autozooids, which have functions related to feeding and reproduction, and siphonozooids for exhaling water from canals, and sometimes other types of zooids (mesozooids, acrozooids) (*Williams, 2011*; *Williams, Hoeksema & van Ofwegen, 2012*). Sea pens have utility in understanding evolution in class Anthozoa as they have unique ecological traits such as unique morphology and differentiation of zooids, active mobile behavior and bioluminescence, all likely stemming from benthic life on sandy bottoms. However, sea pens have been sparsely studied, and basic information on their taxonomy, diversity and phylogeny remains limited (*e.g.*, *Kushida & Reimer, 2019*).

Colony forms and internal morphology, such as types of sclerites and axes, are both important traits for the identification of sea pens. Sea pen colony forms are diverse, and include various shapes such as feather-like, club-like, whip-like, and capitate colonies (*Renilla* species), as well species with a cluster of polyps at the top of the colony (*Umbellula* species) (*Imahara, Iwase & Namikawa, 2014*). Sclerites are small skeletal elements composed of a kind of calcium carbonate ($CaCO_3$), polycrystalline calcite, which aggregates in the soft tissues of most octocorals (*Fabricius & Alderslade, 2001*; *Lau et al., 2020*). The shapes of sea pen sclerites are simpler than those in other octocorals, and include needle, rod, oval, and plate forms. Some sea pens also have sclerites with three longitudinal flanges or warts (*Bayer, Grasshoff & Verseveldt, 1983*; *Williams, 1995*). Sclerites help support colonies as well as act in preventing feeding by predators (*Lewis & von Wallis, 1991*; *West, 1998*). Across evolutionary time, Metazoa have evolved biominerals such as sclerites and spicules at least twenty times, and improving our knowledge of octocoral sclerites can provide relevant information to help understand biomineralization in other taxa (*Knoll, 2003*). Furthermore, information on biomineralization in octocorals is lacking, and thus the accumulation of knowledge is important (*Conci, Vargas & Woerheide, 2021*). Finally, most sea pen taxa except *Renilla*, *Echinoptilum*, *Actinoptilum*, and some *Veretillum*

spp. have a more or less long central axis, or alternately a short axis consisting of calcium carbonate (*Williams, 1995*; *Williams, 2011*).

The traditional classification system of sea pens was primarily established from the end of the 19th century to the early 20th century (*e.g.*, *Kölliker, 1869*; *Kölliker, 1870*; *Kölliker, 1872*; *Kölliker, 1875*; *Kölliker, 1880*; *Kükenthal & Broch, 1911*; *Kükenthal, 1915*; *Hickson, 1916*). *Kükenthal (1915)* developed and established two suborders, Sessiliflorae and Subselliflorae, based on external morphology. However, the validity of these two suborders has been in doubt due to results of cladistic (*Williams, 1995*) and molecular phylogenetic studies, which instead have either shown four large clades within Pennatulacea (*Dolan et al., 2013*; *Kushida & Reimer, 2019*; *García-Cárdenas, Núñez Flores & López-González, 2020*; maximum likelihood analyses), or two clades and an unstable grouping of other taxa (*García-Cárdenas, Núñez Flores & López-González, 2020*; Bayesian analyses). In addition, colony forms of sea pens have been suggested to be the result of convergent evolution (*Dolan et al., 2013*; *Kushida & Reimer, 2019*), making the current taxonomy of Pennatulacea confusing due to inconsistencies between diagnostic morphology and molecular data (*Dolan et al., 2013*; *Kushida & Reimer, 2019*; *García-Cárdenas, Núñez Flores & López-González, 2020*). Additionally, in the case of the axis, most species with no axis are located within a single phylogenetic clade (=clade 2 *sensu Dolan et al. (2013)* and *Kushida & Reimer (2019)*; clade II *sensu García-Cárdenas, Núñez Flores & López-González (2020)*. This clade 2 includes *Renilla* sp., *Echinoptilum* sp. and *Actinoptilum molle* (Kükenthal, 1910). However, some species of *Veretillum* also lack an axis (*Williams, 1995*), but are located in another phylogenetic clade (=clade 1 *sensu Kushida & Reimer (2019)*; clade I *sensu García-Cárdenas, Núñez Flores & López-González (2020)*. Thus, it is highly possible sea pens have lost their axes more than once across their evolution.

Among the taxonomic characters of sea pens, sclerites are an important component, but the evolution of sea pen sclerites has not been studied well. In the case of Caribbean gorgonian octocorals, *Sánchez et al. (2003)* mapped morphological traits such as sclerites and axis mineralization onto the molecular phylogenetic relationships of gorgonians. Their study found the type of surface sclerite present to be very variable, and suggested the diversity of sclerites may be due to their functional morphology, as it is known that sclerite forms can vary in response to the environment (*West, Harvell & Walls, 1993*). *Quattrini et al. (2020)* examined anthozoan skeleton evolution through deep time with regards to ocean conditions with a robust phylogenetic tree. This study noted octocorals have obtained and lost their skeletal forms multiple times across their evolution, and that the skeletal composition of octocorals appears to be able to adapt more readily to changing ocean chemistry when compared to the skeletons of scleractinian hard corals. In addition, these important species may be affected by modern climate change. For example, it is known that higher $pCO_2$ from ocean acidification reduces biomineralization in the red coral *Corallium rubrum* (*Cerrano et al., 2013*; *Rastelli et al., 2020*), and thus climate change may threaten the rich diversity of octocorals.

However, there have only been a few studies on the evolution of sea pens using phylogenetic and morphological information due to extremely limited available genetic information on sea pens (*e.g.*, *Williams, 2011*; *Dolan et al., 2013*; *Kushida & Reimer,*

*2019*; *García-Cárdenas, Núñez Flores & López-González, 2020*). *Kushida & Reimer (2019)* mapped the presence of three-flanged sclerites onto their molecular phylogeny, and found almost all species in clade 1, which includes many shallow water species, lacked three-flanged sclerites, while all taxa in clades 2, 3, and 4 had three-flanged sclerites with the exception of the shallow water species of *Scytalium* (clade 2) and deep-sea *Umbellula* sp. 2 (clade 4, *sensu Dolan et al., 2013*; *Kushida & Reimer, 2019*). In addition, in *Kushida & Reimer's (2020)* formal description of a *Calibelemnon* species, which focused on sea pens within clade 1, they hypothesized that the common ancestor of a molecular *Anthoptilum* –*Umbellula* –*Calibelemnon* clade may have lost their rachis sclerites when ancestral species moved from shallow to deeper waters. Thus, it seems that sclerite evolution may be related to environmental factors such as habitat depth. However, our evolutionary knowledge of sea pens based on direct observations is still fragmentary at best.

Sea pens as a group inhabit a wide range of habitats, living embedded in sandy and muddy bottoms from the intertidal zone to the deep sea down to 6,260 m (*Umbellula*) and from tropical to polar regions (*Williams, 2011*; *Williams, 2021*). Although reports of some sea pen taxa are very limited (*e.g.*, genera *Echinoptilum*, *Calibelemnon*), it is known that 67% of sea pen families (ten of 15 families) and about half of described sea pen genera have a global distribution (*Williams, 2011*). Additionally, instead of regional distributions, it is known that sea pen distributions are apparently based on bathymetric ranges. Accordingly, *Williams (2011)* divided sea pens into three major groups based on their bathymetric ranges: shallow-water species (0–400 m), mid-depth species (400–2,000 m), and deep-water species (2,000–6,260 m). The variability of sea pens and their transition to the deep sea was treated in *Pasternak (1989)*. Similarly, recent work on gorgonians has shown bathymetric gradients act as drivers of diversification (*Sánchez et al., 2021*).

Here, we focused on the sclerites of sea pens, aiming to explore the trends of sclerite adaptation and evolution in relation to the habitat depth of sea pens. We observed sclerites from various species, analyzed their molecular phylogenetic relationships, and overlapped sclerite traits onto a phylogenetic tree to resolve the following questions: (1) are there any relationships between phylogenetic topology and sclerite forms? and (2) are there any relationships between habitat depth and sclerite forms?

## MATERIALS & METHODS

### Specimen sampling

All specimens utilized in this study and their GenBank information (https://www.ncbi. nlm.nih.gov/genbank) are listed in Table S1. In this study, we utilized novel specimens as well as specimens from *Kushida & Reimer (2019)* in our morphological and molecular examinations. Novel specimens were collected from Japanese waters (Table S1) *via* a variety of sampling methods. Mesophotic and deep-sea specimens were collected during the 18th JAMBIO (Japanese Association for Marine Biology) Coastal Organism Joint Survey off Shimoda, Japan in December 2018, T/RV *Seisui-maru* (Mie University) cruise No. 1903 off Kii Peninsula, Japan in April 2019, R/V *Kaimei* cruise KM19-05C (JAMSTEC;

Japan Agency for Marine-Earth Science and Technology) off Iwate, Japan in August 2019, and ROV cruises conducted by Okinawa Churaumi Aquarium off Okinawa, Japan in 2018 to 2019. Additional specimens were collected by reef walking (*Cavernularia* sp.; NSMT-Co1761) and scuba diving (<40 m), as well as by use of dredges, beam trawl nets, an ROV, and a bottom sampler for deeper waters specimens. Furthermore, two preserved specimens from the Western Australian Museum were utilized (Reg. No: WAM Z44543, WAM Z43174), collected during the Antarctic Circumnavigation Expedition 2016–2017. Field surveys requiring permits were conducted under Ministry of Agriculture, Forestry and Fisheries (Directive 30 Sui-kan No. 2709).

It should be noted that although we utilized a relatively comprehensive set of taxa in this study, we were not able to acquire specimens of all genera of sea pens, and there are some taxa missing in this study. In this research, habitat depths were categorized based on *Williams (2011)* and *Kushida & Reimer (2020)* as categories Shallow (0–50 m), Mesophotic (50–200 m), Shallower Deep (200–400 m), Medium Deep (400–2,000 m), and Deep (2,000–6,260 m). Furthermore, we also set broader groups than our initial depth category definitions; a shallower group consisting of the shallow + mesophotic + shallower deep categories (shallower than 400 m; shallow-water species *sensu Williams (2011)*), and a deeper group consisting of the medium deep + deep categories (deeper than 400 m; mid-depth species + deep-water species *sensu Williams (2011)*) for analyses as described below. We divided the depth groups at 400 m based on the distribution patterns of sea pens as described by *Williams (2011)*.

## Morphological observations

The external morphology of specimens was observed with a stereoscopic microscope (Leica S8 APO, Leica Ltd., Tokyo). In this study, surface sclerites such as in the rachis, tentacles, and polyp leaves (if present) were focused on because these parts are considered to be strongly affected by the surrounding environment (*Sánchez et al., 2003*). For sclerite observations, tissues surrounding sclerites were dissolved in sodium hypochlorite (household bleach). After this, the sclerites were rinsed at least three times with water, with a final rinse of ethanol. After aligning isolated sclerites on SEM stubs under a stereomicroscope, sclerites were coated with Pd/Au for observation by a JEOL JSM-6060LV SEM (JEOL, Tokyo, Japan) operated at high vacuum at an acceleration voltage of 15 kV.

## Molecular phylogenetic analyses and trait analyses

The protocols of DNA extraction and amplification in this study were similar to those in *Kushida & Reimer (2019)*. We utilized sequences from the mitochondrial mutS-like protein DNA mismatch repair gene (mtMutS) and the NADH dehydrogenase subunit 2 gene (ND2) as well as sequences of the cytochrome c oxidase I (COI) region following *McFadden et al. (2014)*. To construct robust and comprehensive phylogenetic trees, we additionally utilized published sequences in GenBank from *Dolan et al. (2013)*, *Kushida & Reimer (2019)*; *Kushida & Reimer (2020)*, and *García-Cárdenas, Drewery & López-González (2019)*. These sequences were aligned and concatenated using Geneious v.8.1.3 (*Kearse et al., 2012*). The total length of the resulting alignment was 1812 bp (COI: 651 bp, mtMutS:

696 bp, ND2: 465 bp) and included 74 sequences. Newly acquired sequence information was deposited in NCBI GenBank (National Center for Biotechnology Information, America) (http://www.ncbi.nlm.nih.gov/genbank) (Table S1). As in *Kushida & Reimer (2019)*, sequences from a species of Ellisellidae, sister taxa to sea pens, were utilized as outgroup.

To construct phylogenetic trees, maximum likelihood (ML) and Bayesian (B) analyses were conducted under the general time-reversible model with discrete gamma distribution and invariant sites (GTR+G+I) as suggested by model selection in MEGA X (*Kumar et al., 2018*). Rapid boot strapping and search for best-scoring ML tree were chosen. Bootstrap replicates were performed 1000 times in Geneious v.8.1.3 (*Kearse et al., 2012*). For constructing the Bayesian tree, analyses were conducted in Mr. Bayes v. 3.2.2 (*Huelsenbeck & Ronquist, 2001*; *Ronquist et al., 2012*) under the the Hasegawa-Kishino-Yano model with gamma distribution (HKY+G) model. Markov chain Monte Carlo (MCMC) chains were run for 10,000,000 generations and sampling frequency was set each 200 generations. The first 480,000 generations were discarded as burn-in (average standard deviation of frequency: 0.007987). The average standard deviation of frequency was 0.001552 in the final generation.

The ecological traits of these depth groups and sclerite shapes (no sclerites, plates, smooth rods and needles, three-flanged sclerites, spindles) were mapped onto the ML phylogenetic tree to estimate the ancestral state of sea pen habitat depth. The analyses were performed in Mesquite v. 3.6.1 with maximum likelihood (ML) method with a Mk1 likelihood model (*Maddison & Maddison, 2019*).

The morphological information for specimens with sequences in GenBank was based on *Williams (1995)*, *López-González & Williams (2002)*, *Dolan (2008)*, and *García-Cárdenas, Drewery & López-González (2019)*.

### Statistical analyses

Statistical analyses were conducted in R Console (4.1.0) with RStudio (1.4.1106) User Interface (*R Core Team, 2021*; *R Studio Team, 2021*) to test the relationships among three categories: phylogenetic clades ($n = 4$; clade 1, 2, 3, 4), depth groups ($n = 2$; shallower group, deeper group), and sclerites ($n = 5$; no sclerites, plates, smooth rods and needles, three-flanged sclerites, spindles). Three contingency tables were constructed based on the pairwise comparison of these three categories (clade, depth, sclerite shape). A Pearson's chi-squared test was conducted on each table, and standardized residual *post-hoc* tests with Bonferroni's correction were conducted to detect significant relationships between the categories.

## RESULTS

### Phylogenetic analyses

The ML and B phylogenetic trees of the concatenated mitochondrial COI, mtMutS, and ND2 sequences (Fig. 1) showed a topology divided into four large, well-supported clades; clade 1 (ML bootstrap values/B posterior probabilities = 99/1.00), clade 2 (ML/B = 97/1.00), clade 3 (ML/B = 96/1.00), and clade 4 (ML/B = 86/1.00).

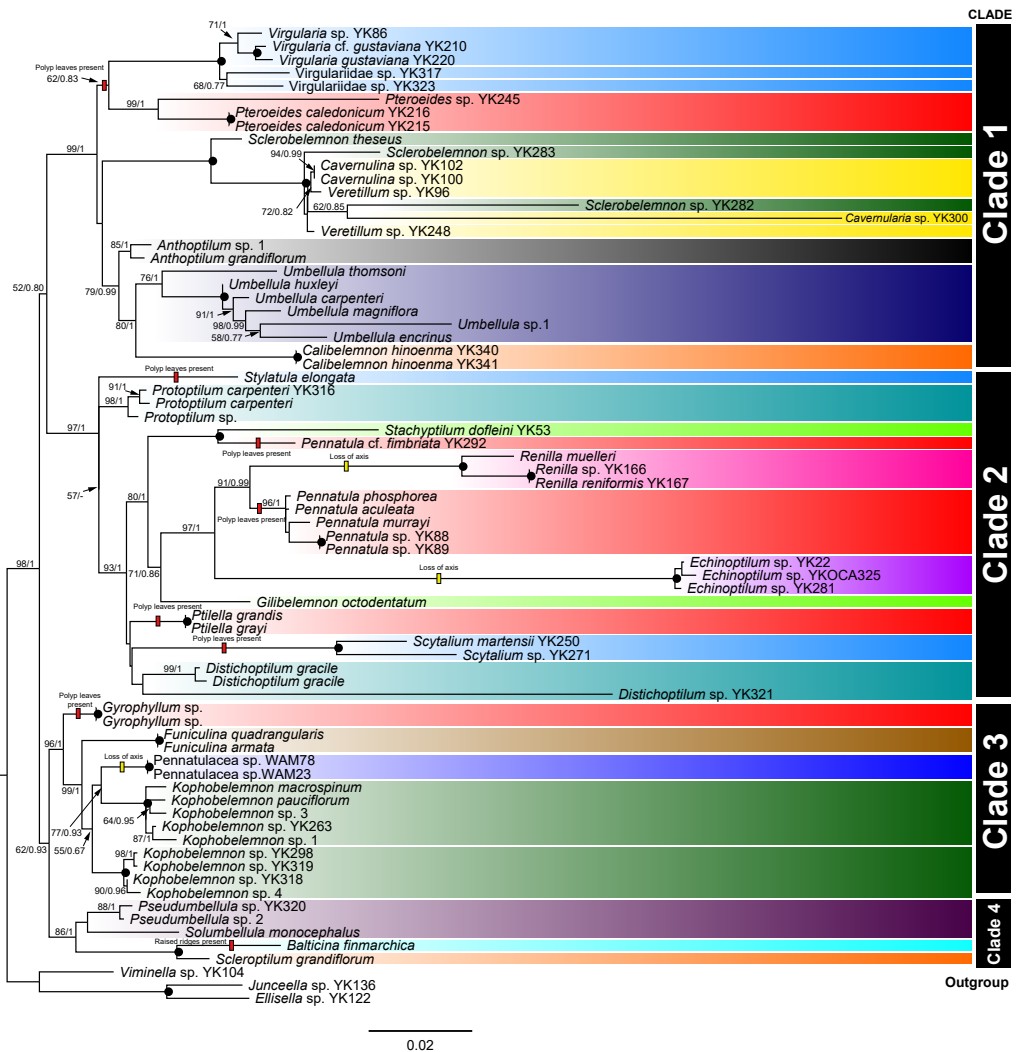

**Figure 1** **COI+mtMutS+ND2 phylogenetic tree constructed by ML method.** The numbers on each branch indicate the ML/B = bootstrap values/posterior probabilities. The black circles on each node indicate the ML/B = 100/1.00. Each color indicates each family; light blue: Virgulariidae, red: Pennatulidae, green: Kophobelemnidae, yellow: Veretillidae, black: Anthoptilidae, dark navy: Umbellulidae, orange: Scleroptilidae, blue green: Protoptilidae, yellow green: Stachyptilidae, pink: Renillidae, light purple: Echinoptilidae, brown: Funiculinidae, Dark purple: Pseudumbellulidae, light blue green: Balticinidae, blue: Pennatulacea sp. The bars at the right side show clades. The small boxes on branches indicate events estimated by maximum parsimony methods; yellow: loss of axis, red: polyp leaves or raised ridges present. Scale indicates substitutions per site.

Clade 1 included a *Virgularia*–Virgulariidae spp.–*Pteroeides* subclade (ML/B = 62/0.83), a *Sclerobelemnon*–Veretillidae subclade (ML/B = 100/1.00), and a *Anthoptilum*–*Umbellula*–*Calibelemnon* subclade (ML/B = 79/0.99). Within the *Virgularia*–Virgulariidae sp.–*Pteroeides* subclade, we observed a small *Virgularia*–Virgulariidae spp. (ML/B = 100/1.00) and *Pteroeides* (ML/B = 99/1.00) clades. Within the *Sclerobelemnon*–Veretillidae subclade (ML/B = 100/1.00), *Veretillum*, *Cavernulina* and *Cavernularia* species were nested with

*Sclerobelemnon* species. Within the *Anthoptilum* –*Umbellula* –*Calibelemnon* subclade, we observed an *Anthoptilum* subclade (ML/B = 85/1.00), and an *Umbellula* –*Calibelemnon* subclade (ML/B = 80/1.00) with *Umbellula* (ML/B = 76/1.00) and *Calibelemnon* (ML/B = 100/1.00) clades.

Clade 2 included a sole representative of *Stylatula elongata* Verrill, 1864, as well as a *Protoptilum* subclade (ML/B = 98/1.00), and a large subclade (ML/B = 93/1.00) that included a *Stachyptilum* –*Pennatula* subclade (ML/B = 100/1.00), a *Renilla* –*Pennatula* –*Echinoptilum* –*Gilibelemnon* subclade (ML/B = 71/0.86), a *Ptilella* clade (ML/B = 100/1.00), a *Scytalium* clade (ML/B = 100/1.00), a *Distichoptilum* clade (ML/B = 99/1.00), and a single *Distichoptilum* representative. Within the *Renilla* –*Pennatula* –*Echinoptilum* –*Gilibelemnon* subclade, there was a *Renilla* –*Pennatula* –*Echinoptilum* clade (ML/B = 97/1.00), with *Renilla* and *Pennatula* as sister taxa (ML/B = 91/0.99).

Clade 3 included a *Gyrophyllum* clade (ML/B = 100/1.00), and a large subclade (ML/B = 99/1.00). This large subclade included a *Funiculina* subclade (ML/B = 100/1.00), and a nested clade (ML/B = 55/0.67) formed of a *Kophobelemnon* subclade (ML/B = 100/1.00) and a *Kophobelemnon* –unidentified specimen subclade (ML/B = 77/0.93) including a *Kophobelemnon* subclade (ML/B = 100/1.00).

Clade 4 included an Pseudumbellulidae clade (ML/B = 88/1.00) and sequences from single representatives of *Balticina finmarchica* (Sars, 1851) and *Scleroptilum grandiflorum* (*Kölliker, 1880*)

Shallow water specimens were included in clades 1 and 2 and a single specimen of shallow water *Funiculina quadrangularis* (Pallas, 1766) was in clade 3. Almost all sea pens from mesophotic depths were included in clade 2, except Pennatulacea sp. in clade 3. Shallower deep specimens were located in clades 1, 2 and 3. Medium deep and deep water specimens were located in all clades. Each clade included specimens of various colony shapes (Figs. 2–4).

The results of the ML ancestral reconstruction (Fig. 5) suggested the last common ancestor of all sea pens was likely from deeper than 400 m (proportional likelihood = 0.95). The habitat depths of each common ancestor of clade 1, clade 2, clade 3 and clade 4 were also deeper than 400 m (proportional likelihoods: clade 1 = 0.82, clade 2 = 0.90, clade 3 = 0.99, clade 4 = 0.99). Furthermore, the sclerite shape of the common ancestor of all sea pens was likely the three-flanged sclerite (proportional likelihood = 0.99). The common ancestor of clade 1 had lost sclerites (proportional likelihood = 0.94), and subsequently several taxa re-obtained sclerites such as plates, smooth rods or needles, while only the deep-sea species *Umbellula thomsoni* (*Kölliker, 1874*) re-obtained three-flanged sclerites. In addition, the shapes of sclerites of the common ancestors of other clades were also likely three-flanged structures (proportional likelihoods: clade 2 = 0.99, clade 3 = 0.99, clade 4 = 0.99). Subsequently, *Scytalium* species evolved plate sclerites in shallow water, while Pennatulacea sp. (WAM Z44543) and *Solumbellula monocephalus* Pasternak, 1964 developed spindles (Fig. 5).

 

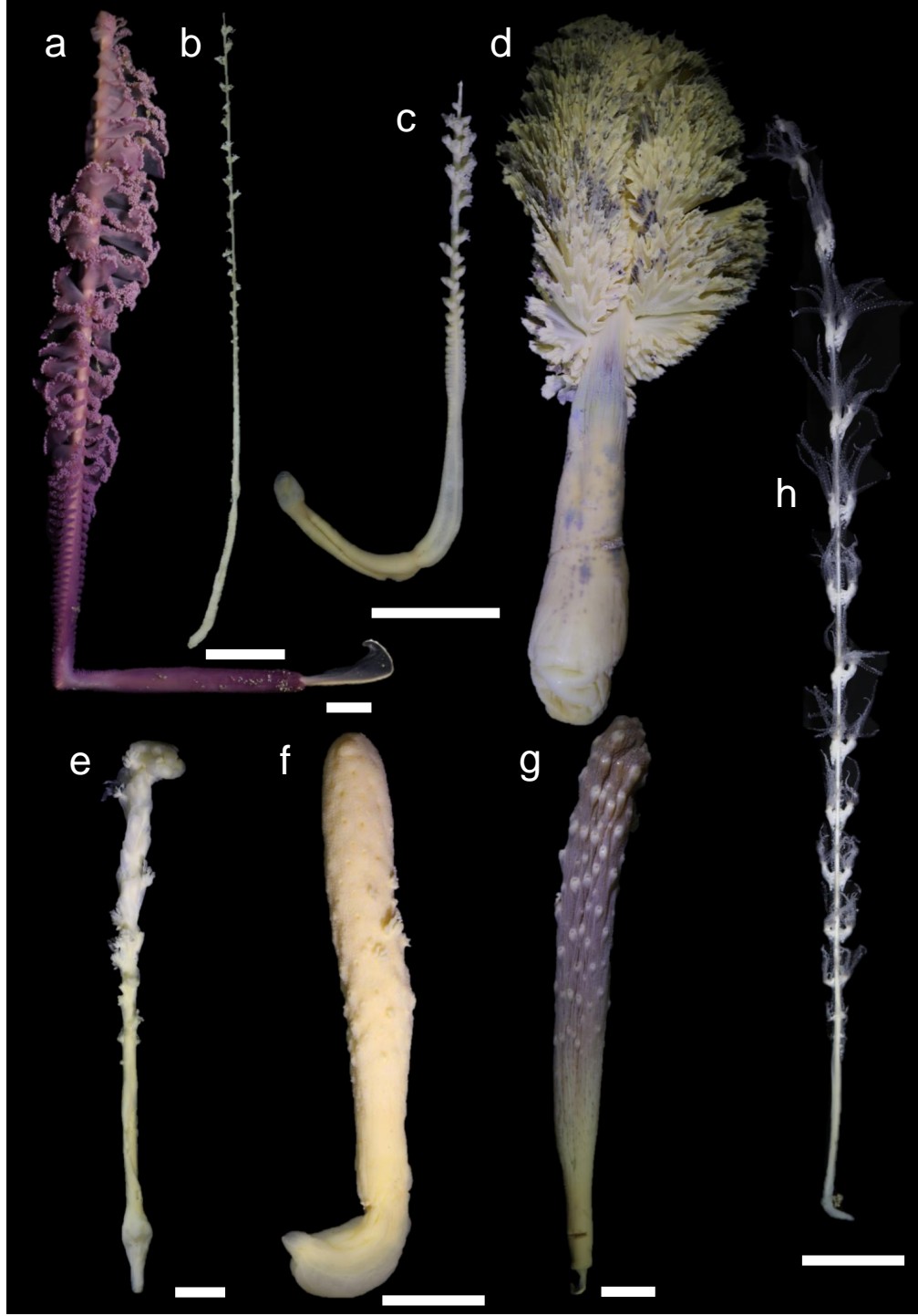

**Figure 2 Specimens in clade 1.** (A): living specimen (B-H): preserved specimens in 99.5% ethanol. (A): *Virgularia* cf. *gustaviana* (NSMT-Co 1776), 9 m in depth, (B): Virgulariidae sp. (NSMT-Co 1779) 609-612 m, (C): Virgulariidae sp. (NSMT-Co 1780) 400 m, (D): *Pteroeides caledonicum* (NSMT-Co1773), 10 m, (E): *Sclerobelemnon* sp. (NSMT-Co 1765), 207 m, (F): *Cavernulina* sp. (NSMT-Co1763), 20 m, (G): *Veretillum* sp. (NSMT-Co1762), 17 m, (H): *Calibelemnon hinoenma* (NSMT-Co 1709), 32 m. Scale bars = one cm.

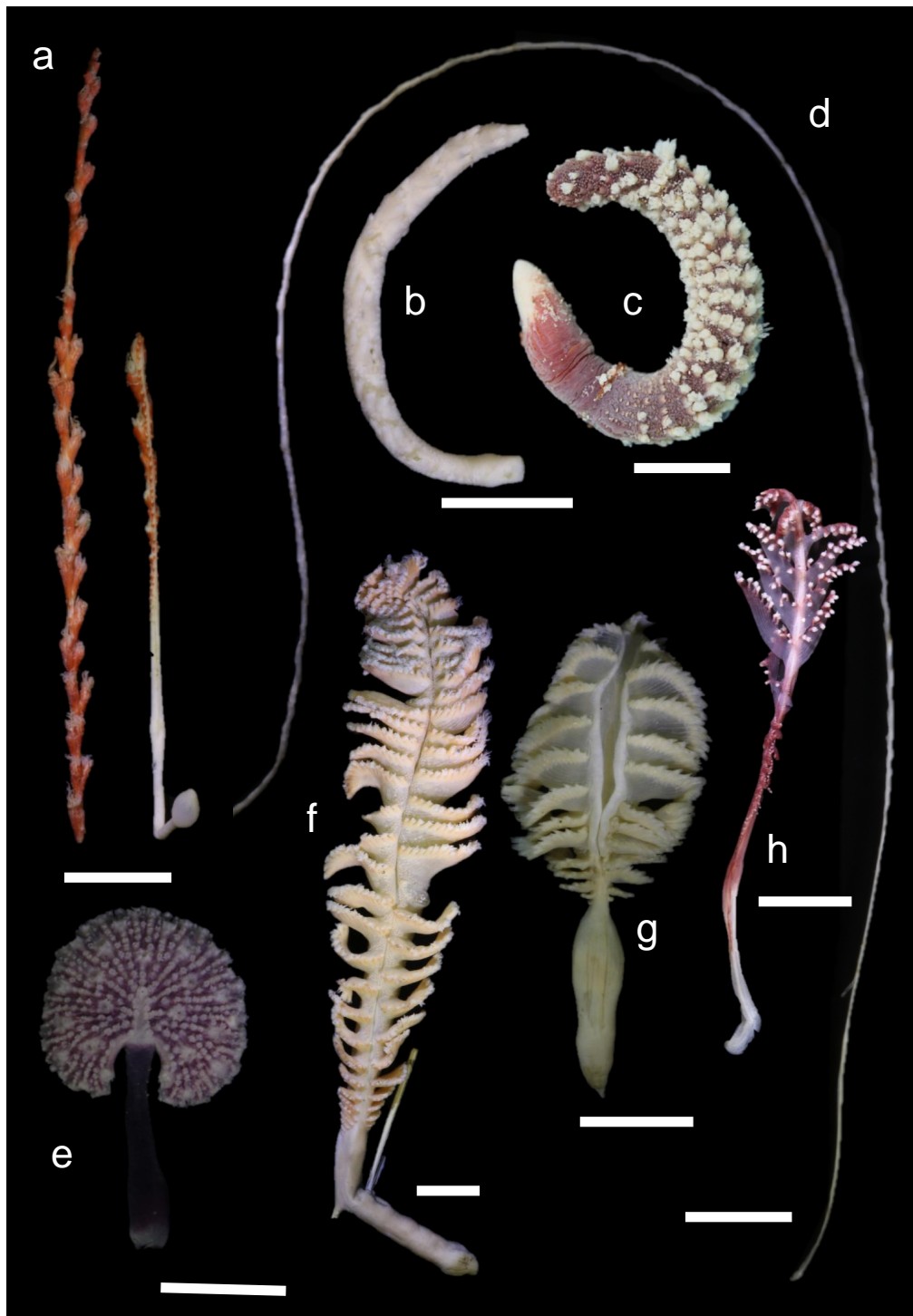

**Figure 3   Specimens in clade 2.** (A) *Protoptilum carpenteri* (NSMT-Co1771), 609–612 m in depth, (B): *Stachyptilum dofleini* (NSMT-Co1769), 95–187 m, (C): *Echinoptilum* sp. (NSMT-Co1764), 118 m, (D): *Distichoptilum* sp. (NSMT-Co1770), 694-752 m, (E): *Renilla reniformis* (NSMT-Co1760), 5 m, (F): *Pennatula* sp. (NSMT-Co1774), 77-102 m, (G): *Pennatula* cf. *fimbriata* (YK292), 190 m, (H): *Scytalium* sp. (NSMT-Co1778), 22 m. Scale bars = one cm. All preserved specimens in 99.5% ethanol.

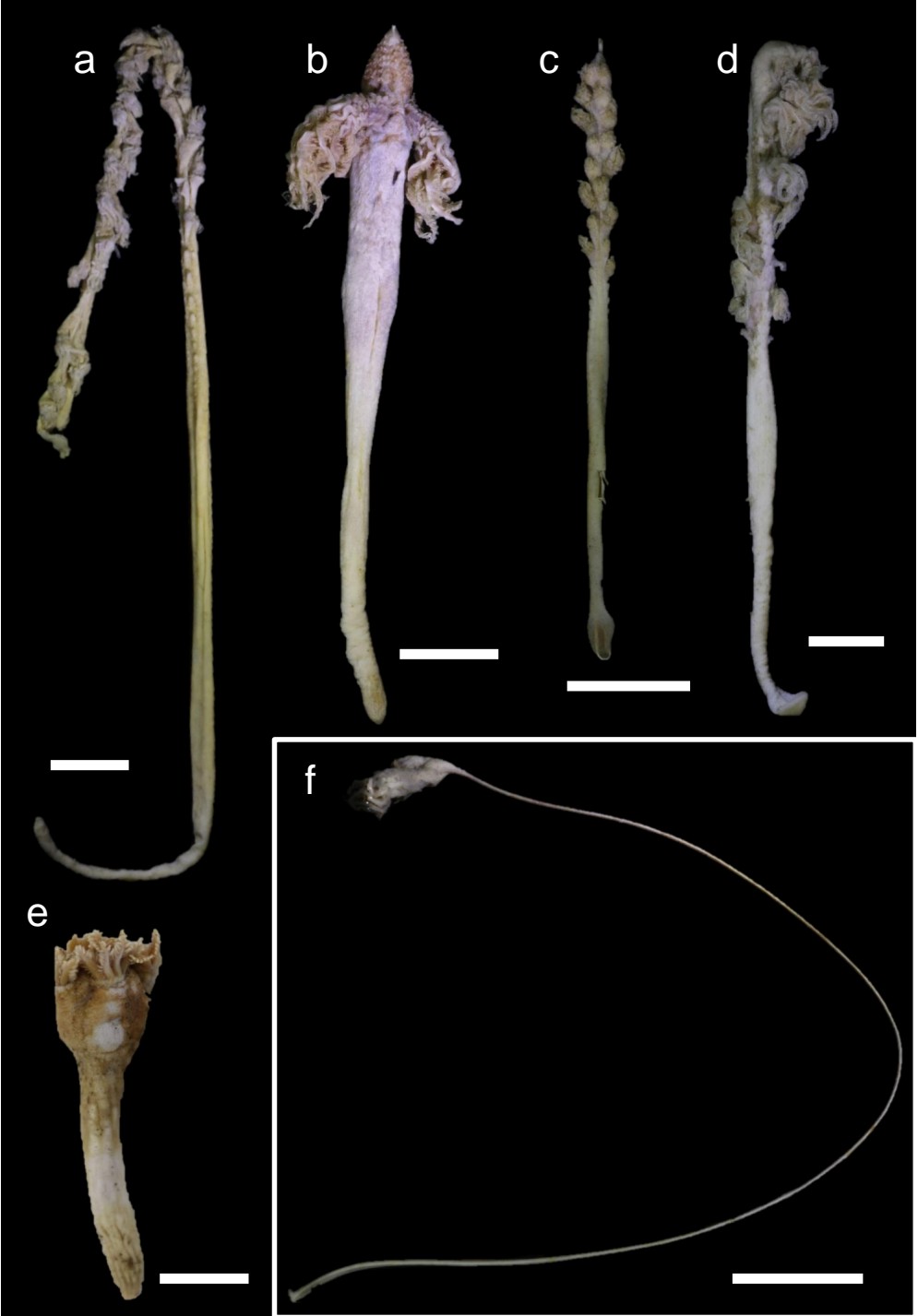

**Figure 4** **Specimens in (A–E): clades 3 and f: clade 4.** (A) *Kophobelemnon* sp. (NSMT-Co 1768), 694–752 m in depth, (B): *Kophobelemnon* sp. (NSMT-Co 1767), 694–752 m, (C): *Kophobelemnon* sp. (NSMT-Co1766), 400–450 m, (D): *Kophobelemnon* sp. (YK263), 729–793 m, (E): Pennatulacea sp. (WAMZ44543), 185 m, (F): *Pseudumbellula* sp. (NSMT-Co1772), 694–752 m. Scale bars = one cm. All preserved specimens in 99.5% ethanol.

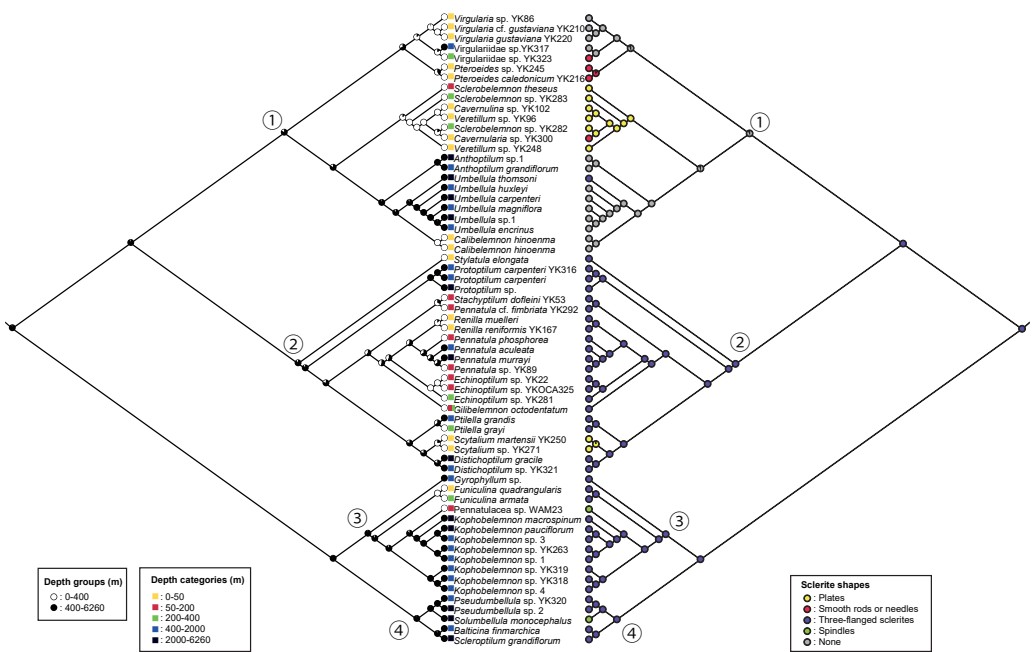

**Figure 5** **Ancestral state reconstruction for habitat depth by ML method.** Monochrome circle color in the left tree indicates each depth group; white: the shallower group (0–400 m), black: the deeper group (400–6260 m) and each box's color each depth categories; yellow: shallow (0–50 m), red: mesophotic (50–200 m), green: shallower deep (200–400 m), blue: medium deep (400–2000 m), and black: deep (2000–6260 m). Circle colors in the right tree indicate each shape of sclerites in the rachis; gray: none, yellow: plates, red: smooth rods or needles, blue: three-flanged sclerites, yellow green: spindles. Each number in a circle indicates each clade.

## Sclerite observations

The results of sclerite observations from representative specimens are listed in Table S2. Regarding trends of sclerite forms of each clade, species in clade 1 exclusively had plates, smooth rods or smooth needle sclerites, or no sclerites at all (Fig. S1), while species in clade 2 had three-flanged rods or needles as well as plates for shallow water species of *Scytalium* (Fig. S2). Species in clade 3 also had three-flanged rods or needles or spindles with warts (Fig. S3). Due to the limited number of specimens in clade 4, we could only examine a single specimen of *Pseudumbellula* sp. (NSMT-Co1772). This *Pseudumbellula* specimen had several different types of sclerites, including rods with spines, spindle, three-flanged rods in tentacles, and three-flanged rods and three-flanged spindles with serrated edges in the rachis (Fig. 6). In addition, broken three-flanged sclerites were observed in multiple clades in this study. These damaged sclerites had broken flanged portions with an undamaged central part (Fig. 7).

## Statistical analyses

Depth preferences among clades (Table S3) showed species from shallower depths were higher in number in clades 1 ($n = 15$) and 2 ($n = 13$), and rarely from clades 3 ($n = 3$), with none present in clade 4 ($n = 0$). On the other hand, clades 1 to 3 had equal numbers

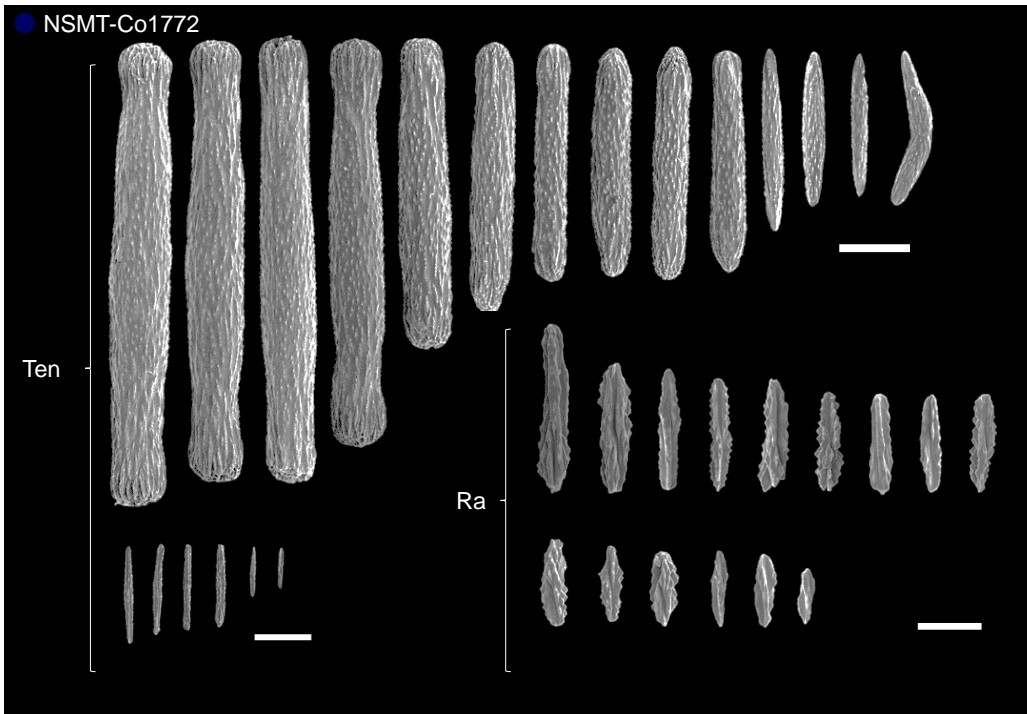

**Figure 6  Sclerites in clade 4 (single specimen *Pseudumbellula* sp. (NSMT-Co1772)).** Abbreviations; Ten, tentacles; Ra, rachis. Scale bars = Ten: 200 μm, Rachis = 100 μm.

of species ($n = 9$) at deeper depths, and clade 4 included only deeper species ($n = 5$). There was a significant difference among the depth groups with clades ($X^2 = 10.21$, $p = 0.009$). However, the standardized residual (|2.73|) did not detect any significant relationship between each depth group with clade.

Comparisons between clades and sclerites (Table S4) showed the majority of species from clade 1 ($n = 13$) did not possess sclerites, and the other clade 1 species contained exclusively plates ($n = 6$), or smooth rods or needles ($n = 4$). On the other hand, clades 2, 3 and 4 had mostly three-flanged sclerites. There was a significant difference among sclerite shapes in the rachis by clade ($X^2 = 58.93$, $p = 0.0004$). The standardized residual (|3.02|) test showed the consistency of no sclerites and three flanged sclerites found in clade 1 and of three-flanged sclerites in clade 2.

Furthermore, plates and smooth rods or needles were found only in the shallower group (Table S5). Three-flanged sclerites, spindles and species with no sclerites were observed in both depth groups. There was a significant difference between the two depth groups with regards to sclerite types present ($X^2 = 15.45$, $p = 0.0009$). In addition, plates were significantly more likely to be present in the shallower group and significantly less likely to be in the deeper group based on standardized residual test with Bonferroni's correction (|2.80|). Of note, when we conducted supplemental statistical analyses between clades and sclerite forms by uniting plates and smooth rods or needles as "simple surface sclerites",

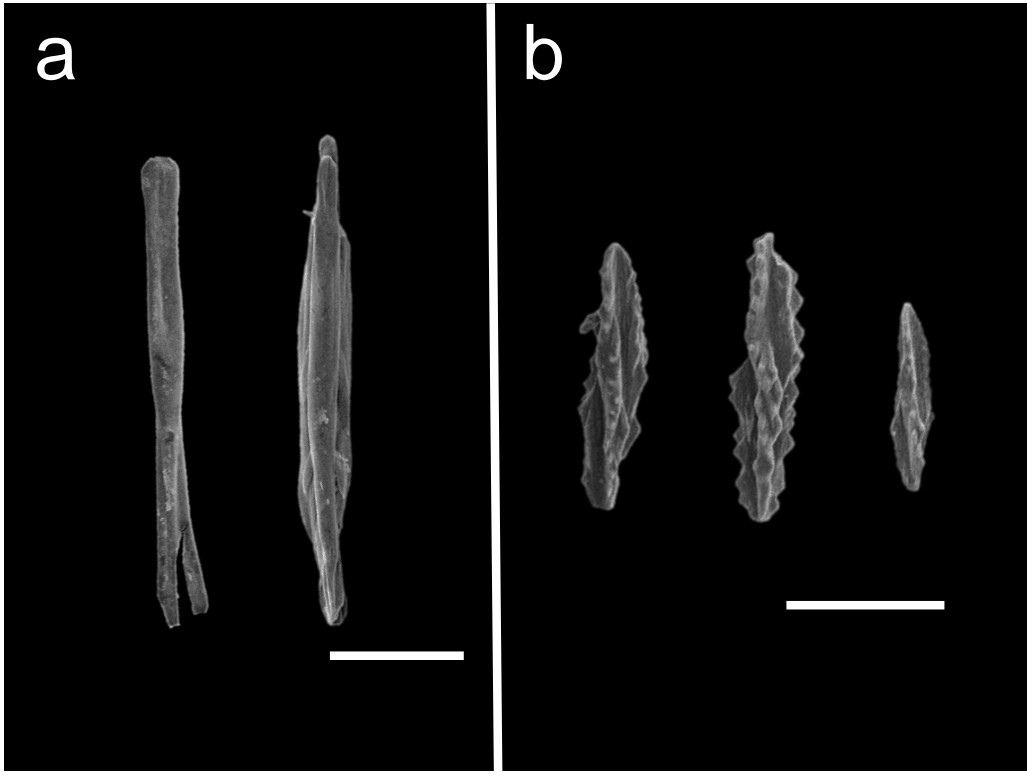

**Figure 7** **Broken three-flanged sclerites of different clades.** (A): NSMT-Co1774 in clade 2, polyp leaves, (B): NSMT-Co1772 in clade 4, rachis. Scale bar = 100 μm.

significant relationships between clades 1 and 2 –none, clade1 –simple surface sclerites, and clades 1 and 2 –three-flanged sclerites were detected (|2.95|).

## DISCUSSION

### Unexpected diversity of sea pens

During this study, we examined several remarkable specimens. Among these was a specimen of Virgulariidae sp. (NSMT-Co1780) that was collected off Ofunato, Iwate, Japan, at a depth of 400 m (Fig. 2C). This specimen had thick polyp leaves and thin rod sclerites that were intermediate between plates and rods in their tentacles (Fig. S1). At first, this specimen was thought to be a species of *Scytalium*, which have thin polyp leaves and plate-like thin rod sclerites, however, this specimen's phylogenetic position was closer to *Virgularia* species rather than to *Scytalium* species. Therefore, based on its phylogenetic position, it is possible that this species belongs to an undescribed genus.

Another Virgulariidae sp. specimen (NSMT-Co1779), collected from Tanabe Bay, Wakayama, Japan, at depths between 609 and 612 m (Fig. 2B), was similar to *Virgularia* and *Calibelemnon* species as it did not have sclerites in its rachis and polyps. At first observation this specimen appeared to belong to *Calibelemnon* (*Kimura et al., 2019*) as the polyps were paired and there were no terminal polyps. However, this specimen was

phylogenetically located as a sister taxon to Virgulariidae sp. (NSMT-Co 1780), and was not close to *Calibelemnon* specimens. Thus, it is possible that this specimen represents another undescribed genus within Virgulariidae.

Finally, two specimens of an unknown taxa of Pennatulacea sp. (WAM Z44543, Z43174) from the Western Australia Museum collection (Fig. 4E), which were collected from sub-Antarctic Candlemas Island at a depth of 185 m, were unusual, as their colony forms were intermediate between *Kophobelemnon* and *Umbellula* species. However, these specimens did not have axes, and had spindle-like sclerites. Based on these two specimens' phylogenetic position within clade 3, it is possible that these specimens belong to an undescribed family, or alternately that there is a need to reconsider the taxonomic definition of family Kophobelemnidae.

In the past two decades, novel sea pen genera have been reported from the Antarctica region and from the deep sea (*e.g.*, *López-González & Williams, 2002*; *López-González, Drewery & Collins, 2022*). These previous findings and our current results suggest that the diversity of sea pens likely exceeds our current understanding, and that further surveys both of shallow waters (*Kushida & Reimer, 2019*) and the deep sea are needed. Further specimen collections from a wide range of locations should help to resolve knowledge gaps currently hampering sea pen research.

## Phylogenetic relationships among taxa

The general topology of our tree supported the results of *Dolan et al. (2013)*, *Kushida & Reimer (2019)*, and the ML tree in *García-Cárdenas, Núñez Flores & López-González (2020)*. *García-Cárdenas, Núñez Flores & López-González (2020)*, who recovered different results based on their ML and Bayesian analyses, mentioned the possibility of bias from using only mtDNA sequences, and their 28S rDNA sequences appear useful in making the support of clades 1 and 2 stronger, and in demonstrating the unstable status of clades 3 and 4 in their Bayesian tree. However, the utility of the combination of mtMutS and ND2 for genus- or family-level analyses has been supported by previous studies (*e.g.*, *McFadden et al., 2006*; *Quattrini et al., 2020*). As well, the unstable lower support values of clades 3 and 4 in the Bayesian tree in *García-Cárdenas, Núñez Flores & López-González (2020)* could be related to the limited taxon sampling in their study. Alternately, the lower support values of clades 3 and 4 may be caused by past patterns of diversification and extinction (*i.e.*, hard polytomy) rather than data limitations (*i.e.*, soft polytomy). Further verification *via* more robust next generation analyses as in *Quattrini et al. (2020)* and *McFadden et al. (2021)* may help to resolve this issue.

So far, at least seven families of sea pens (Virgulariidae, Pennatulidae, Scleroptilidae, Kophobelemnidae, Protoptilidae, Stachyptilidae, Umbellulidae) are known as polyphyletic based on the molecular results of precious studies *e.g.*, *Dolan et al., 2013*; *Kushida & Reimer, 2019*; and the results of both of these past studies were supported by the current study. The original family Umbellulidae was recently separated to two families, Umbellulidae and Pseudumbellulidae (*López-González, Drewery & Collins, 2022*), and the current study is the first report of Pseudumbellulidae species from the northwestern Pacific Ocean (Japan), with specimens from the shallowest depths reported thus far; 694–752 m (*Pseudumbellula*

sp. NSMT-Co1772). The genus *Kophobelemnon* may also be rendered paraphyletic if unknown Pennatulacea sp. specimens (WAM Z44543, WAM Z43174) are not assigned to an expanded-concept *Kophobelemnon*. The genus *Pennatula* was also recovered as polyphyletic, with species located in the *Stachyptilum –Pennatula* subclade (ML/B = 100/1) and in the *Renilla –Pennatula –Echinoptilum –Gilibelemnon* subclade (ML/B = 71/0.86) within clade 2. The genera *Ptilella* and *Alloptilella* were recently resurrected and established as separate from *Pennatula* by *García-Cárdenas, Drewery & López-González (2019)* and *Li, Zhan & Xu (2021)*, respectively, based on the polyphyly of genus *Pennatula*. As polyp leaves have appeared more than once due to convergent evolution in Subselliflorae, it is not surprising that additional taxa in subclass Subselliflorae, such as *Pennatula*, have turned out to be polyphyletic.

## Adaptation and evolution of sea pens across habitat depths

The community structure of octocorals and sea pens is known to be strongly related to bathymetry (*Braga-Henriques et al., 2013*; *Williams, 1992*; *Williams, 2011*). Furthermore, octocorals are suggested to be distributed by their habitat depths to avoid overlap between closely related taxa (*Pante et al., 2012*). Although habitat depth seems to play an important role in the divergence and evolution of octocorals, the divergence and evolution patterns of sea pens along depth have not been well studied to date (*Dolan et al., 2013*; *Kushida & Reimer, 2019*; *Hogan et al., 2019*; *García-Cárdenas, Drewery & López-González, 2019*; *García-Cárdenas, Núñez Flores & López-González, 2020*).

In this study, we overlaid our observations of sclerites in the rachis from representative sea pen specimens onto the phylogenetic topologies within Pennatulacea, and then observed trends of adaptation and evolution of sea pens in each clade, considering colony forms, axis, and sclerites. Our results reconfirmed that colony forms of sea pens have undergone convergent evolution as previously suggested by *Dolan et al. (2013)* and *Kushida & Reimer (2019)*. As polyp leaves have evolved independently multiple times in the Subselliflorae, we hypothesize that these structures allow effective feeding in the water column above soft sediments. Additionally, the loss of axis has independently evolved in *Renilla* species (in clade 2), *Echinoptilum* (in clade 2), and Pennatulacea sp. (in clade 3) (Fig. 1).

Our results demonstrate that there are trends in habitat depths for sea pens in different clades, and thus that the divergence of sea pens may depend at least partially on habitat depth (Fig. 5). Shallow water species (0–50 m) were exclusively located in clades 1 and 2 in addition to a sole species of *F. quadrangularis* (Pallas, 1766) within clade 3, while deep-sea species were present in all clades. As well, most mesophotic sea pens were included in clade 2. Although standardized residual analyses did not detect significant relationships between each depth group with clade, this may be due to the presence of only a single *Anthoptilum –Umbellula –Calibelemnon* subclade, which includes most all deep-sea taxa, in clade 1. Thus, undersampling of some habitats may still be affecting analyses. Furthermore, almost all members of clade 3 and 4 were from deep sea.

## The common ancestor of sea pens

*Pseudumbellula* sp. (NSMT-Co1772) had rods with spines, spindles, and three-flanged rods in tentacles, and three-flanged rods and three-flanged spindles with serrated edges in

the rachis. The sclerite shapes of this specimen, particularly rods with spines and spindles in their tentacles and the serrated edge structures in the rachis, resemble the sclerites of alcyonacean species (*e.g.*, see family Ellisellidae section in *Fabricius & Alderslade (2001)*). *Dolan et al. (2013)* considered that deep-sea taxa may be the origin of sea pens. The current study hypothesizes that the ancestral sea pen in the deep sea had an axis, bilateral traits, three-flanged sclerites, and no calyces (see also *Kushida & Reimer, 2019*). These characters agree well with the morphology of species of Umbellulidae and/or Pseudumbellulidae. From the findings of the current study, it is possible that the common ancestor of sea pens was similar to Pseudumbellulidae, which is a deep-sea taxon, having sclerites intermediate in shape between alcyonaceans' spindles and sea pens' three-flanged sclerites, as well as an axis, and bilateral traits.

Interestingly, some keratoisidid sclerites resembled those of *Pseudumbellula* (both rods with spines and a sclerite structure with serrated edges) (*Watling & France, 2011*; *Dueñas, Alderslade & Sánchez, 2014*). Even though Kerastoisididae is not a sister taxa of Pennatulacea, it seems these sclerite shapes have evolved independently in Octocorallia (*Saucier, France & Watling, 2021*).

## Evolution of sea-pen sclerites across habitat depths

We observed relationships of sea pen sclerite forms with clades and habitat depths, and thus consider that the divergence of sclerite forms of sea pens may also depend on habitat depth (Table S4, Fig. 5). Sclerites shapes in the rachis are significantly diverse, especially in clade 1. Species in clade 1 have smooth rods, needles, plates, or no sclerites in shallower species, while the deeper species we examined lacked sclerites except for *Umbellula thomsoni* (Kolliker, 1874), which had three-flanged sclerites. Almost all species in clades 2, 3, and 4 exclusively had three-flanged sclerites.

From the results of sclerites observation and molecular data, we propose a possible scenario of sclerite evolution: (1) the common deep-sea ancestor of sea pens included intermediate sclerites between alcyonacean spindles and sea pen three-flanged sclerites (as in Fig. 6), (2) sea pen sclerites became simplified by losing serrated edges and three-flanged sclerites became dominant, and (3) various sclerite forms (plates, smooth rods and needles, spindles and no sclerites) appeared (Fig. 5). Furthermore, we confirmed the characteristic evolutionary pattern of the appearance of plates and smooth rods and needles when sea pens advance to shallow waters. Our study also supports the hypothesis of *Quattrini et al. (2020)*, who suggested that octocoral skeletal structures can easily adapt to their surrounding environment. From the results of *Quattrini et al. (2020)* and this study, sea pen species that have specific types of sclerites may have adapted to their surrounding environments.

In the current study, three-flanged sclerites were predicted to have been present in the deep-sea common ancestor of sea pens, and were confirmed in species from various depths in all clades. The reason why sea pens produce three-flanged sclerites remains unknown. We believe there may be a relationship with at least these factors: (1) evolutionary background, and (2) water temperatures. Similarly, it has previously been theorized that the surrounding

environment may affect the primary crystallization process that forms sclerites (*Weiner & Addadi, 2011*).

Firstly, producing three-flanged sclerites may be associated with a species' evolutionary background. The presence of this type of sclerite is suggested to be a relatively conservative trait that originated in the deep sea (Fig. 5). Thus, perhaps many sea pens produce three-flanged sclerites simply as a preserved character from their evolutionary past.

Another possible factor may be the surrounding chemical traits of the marine environment (*Quattrini et al., 2020*). For example, water temperatures may also be associated with the process of sclerite production. From research on scleractinians, it is known that biomineralization is affected by water temperatures as well as Mg/Ca ratios. When the Mg/Ca ratio is low, growth rates of scleractinian skeletons are faster at high water temperatures and slower at low temperatures (*Higuchi et al., 2017*). In the deep sea, it is known that the amounts of water temperature changes are different at different depths. In particular, a thermocline where the water temperature changes widely generally exists at depths from hundreds of meters to 1,000 m. In this study, sea pen species below 400 m that have sclerites were shown to exclusively have three-flanged sclerites. Thus, sclerite shape may possibly be affected by some kind of environmental restriction, and it may be that simple and robust sclerites such as smooth rods or needles are comparatively difficult to produce in low temperature environments. To verify this hypothesis, further molecular ecological experiments are needed.

In addition, almost all specimens from below 400 m in this study had three-flanged sclerites, if sclerites were present in the rachis (Fig. 5). Additionally, in clade 2, *Pennatula* sp. (NSMT-Co1774) from mesophotic depths (77–102 m) and *Renilla* sp. from shallow water (5 m) also had three-flanged sclerites. Interestingly, the three-flanged structures of these two specimens were not as distinct as those of species living in deeper waters. However, in this research, we could not examine specimens of sea pen genera with three-flanged sclerites that inhabit <400 m in depth, such as *Actinoptilum*, *Ptilosarcus*, *Grasshoffia*, and *Acanthoptilum*. Increasing the number of examined taxa will help improve our understanding of these sclerites and their evolution and importance.

The structural forms of biominerals that result from their production process are known to help in damage tolerance in other marine invertebrates (*Yang et al., 2022*). In sea pens, we hypothesize that three-flanged sclerites are possibly adaptations to deep-sea conditions. Our observations of broken three-flanged sclerites support this hypothesis, as in such damaged sclerites the main central parts were still intact and only the flanged parts were broken (Fig. 7). Even if sclerites are chemically vulnerable, and despite strong physical pressure, three-flanged sclerites' function in structural colony support can be maintained with the loss of the flanged part of a sclerite. For similar reasons, serrated edge ornamentation may also be an adapted trait to protect chemically vulnerable sclerites. We observed damaged sclerites in deep-sea *Pseudumbellula* specimens that had undamaged main central parts and only damaged serrated edge ornamentations (Fig. 7). The flanges and serrated-edges may be the sclerite parts that are initially broken while protecting the main central parts and the overall sclerite function. Thus, sea pens with three-flanged

sclerites may be able to better withstand deep-sea environmental pressures without making costly huge robust sclerites.

The acquisition of three-flanged sclerites may also affect sea pens' ecological traits. *Kushida, Higashiji & Reimer (2020)* reported on mole-like burrowing behavior and bending of *Echinoptilum* sp. emerging from within sand, and these behaviors may exert physical pressure on sea pen colonies. *Echinoptilum* sp. may be enabled to perform such behaviors due to the presence of three-flanged sclerites (*Kushida, Higashiji & Reimer, 2020*). Even if there are strong physical pressures *via* moving within the sand, the main central part of three-flanged sclerites remains.

In the cases of other marine animals, sponges (Porifera) and echinoderms (Echinodermata) also have calcareous small elements of $CaCO_3$ (spicules) embedded within their tissues. The shapes of spicules in these marine invertebrates are different from those found in sea pen sclerites. However, it is known that concentrical ultrastructural layers in spicules can be found across taxa (*e.g.*, calcareous sponges and sea urchins; *Sethmann & Wörheide, 2008*). In addition, these structures make spicules strong against fracture stress as the structure can distribute physical stress and deflect propagating fractures (*Sethmann & Wörheide, 2008*). Although our current study did not include the observation of the ultrastructure of sclerites in sea pens, and thus our examinations are from an external point of view, we believe it to be possible that three-flanged structures in sea pens have tolerance towards physical pressures for similar reasons as observed in sponges and echinoderms.

In contrast, plate sclerites are suggested to be a trait adapted towards shallow waters (<400 m), as plates were commonly observed in shallow water species in significantly higher numbers (Table S5). As well, simple surface sclerites (plates + smooth rods or needles) were also shown to be significantly more common in clade 1, which includes many shallow water species, and were also seen in shallow-water *Scytalium* species in clade 2 (plates). Thin plate sclerites without ornamentation may cost less energy to produce compared to three-flanged sclerites (*Kushida & Reimer, 2020*). On the other hand, the supersaturated situation in shallow water may be related to the production of robust smooth rods and needles in shallow water sea pens.

Furthermore, sea pen species with no sclerites in the rachis are thus far only known from clade 1 (*Kushida & Reimer, 2020*). In the taxa utilized in the molecular phylogenetic analyses, the condition of having no sclerites in the rachis has been reported from *Virgularia* and *Calibelemnon* species, and from almost all *Umbellula* species except *Umbellula thomsoni* from *Dolan (2008)* and *Anthoptilum* species in clade 1 (*Williams, 1995*; *Kushida & Reimer, 2020*). Thus, the condition of having no sclerites is suggested to be a derived trait of sea pens. This is consistent with the suggestion of *Quattrini et al. (2020)* that loss of sclerites is most likely a derived condition. It is known that some genera such as *Porcupinella*, *Chunella* and *Amphiacme* in family Chunellidae that inhabit depths between 1,000 and 6,000 m do not have sclerites (*Williams, 1990*; *Williams, 2021*). However, these taxa have not yet been phylogenetically examined, and this information is needed to resolve not only the evolution of sclerite loss, but also to help resolve taxonomic problems between Chunellidae and Scleroptilidae (*Kushida & Reimer, 2020*; *Williams, 2021*). A lack of sclerites in the rachis has been observed in species living across a wide range of depths (shallow, medium deep

and deep). It is also known that several other octocoral taxa do not have sclerites (*Fabricius & Aldersland, 2001*; *Lau & Reimer, 2019*). Similarly, *Virgularia* aff. *gustaviana*, which also has no sclerites, is an ecosystem engineer forming sea pen fields with remarkably high densities (*Kushida et al., 2020*). By not producing sclerites and saving energy, faster growth rates may increase colony numbers in sandy or muddy bottoms, although there may be higher predation risks due to the lack of sclerites. We expect that sclerite types can affect a species' ecological traits. Furthermore, research into sea pen species that lack sclerites may provide additional information in resolving key questions of biomineralization.

The lack of sclerites in polyp tentacles has been observed across a wide range of sea pen taxa. Thus, the presence or absence of tentacle sclerites may not be related to sea pens' phylogenetic relationships or to habitat depths. However, the shapes of sclerites in tentacles are considered to be related to habitat depth, as they are basically similar to the structures of sclerites in the rachis.

Finally, sclerites embedded in sea pens are relatively more simple and smoother than those observed in many other octocoral species. Among octocorals, alcyonaceans spend their life attached to substrates and do not move from their habitat, and prevent predation from other marine animals with their sclerites. However, many sea pens can actively move, including withdrawal behavior and mole-like burrowing (in *Echinoptilum*), and thus apparently at least some sea pens can escape from predators *via* such movements (*Imahara & Ogawa, 2006*; *Ambroso et al., 2013*; *Chimienti, Angeletti & Mastrototaro, 2018*; *Kushida, Higashiji & Reimer, 2020*). Therefore, sea pens may not need the complex and uneven protective sclerites seen in alcyonacean species due to their ability to move.

## CONCLUSIONS

This study increases our knowledge on the phylogeny, adaptation, evolutionary history, and diversity of sea pens. Our results suggest that sea pen diversification is related to habitat depth (Fig. 8), and that sclerite shape may be related to adaptation towards the surrounding seawater chemical environment, and specifically to water temperatures. The common ancestor of sea pens is estimated to have had three-flanged structures and have lived in the deep sea. As sea pens diversified into shallower water, they may have changed the characters of their sclerites, obtaining plates, smooth rods or needles, or even losing sclerites. In addition, sea pen colony forms have experienced convergent evolution in soft sediment environments, and the axis of some sea pen species has been independently lost. Furthermore, we hypothesize that the shapes of sclerites in the rachis are associated to the ecological traits of each sea pen species, such as movement or the ability to form sea pen fields. Recently, the ecological importance of sea pens has been recognized. For example, sea pen fields have been designated as Vulnerable Marine Ecosystems (VMEs) by the Northwest Atlantic Fisheries Organization NAFO and as "Threatened and/or Declining Habitat" by the Oslo and Paris Conventions (OSPAR) (*e.g.*, *Kushida et al., 2020*). As the biomineralization of octocorals is an important factor in create their biodiversity, understanding the mechanisms of sea pens' ecological traits, including their roles as ecosystem engineers *via* more basic and ecological studies are urgently needed.

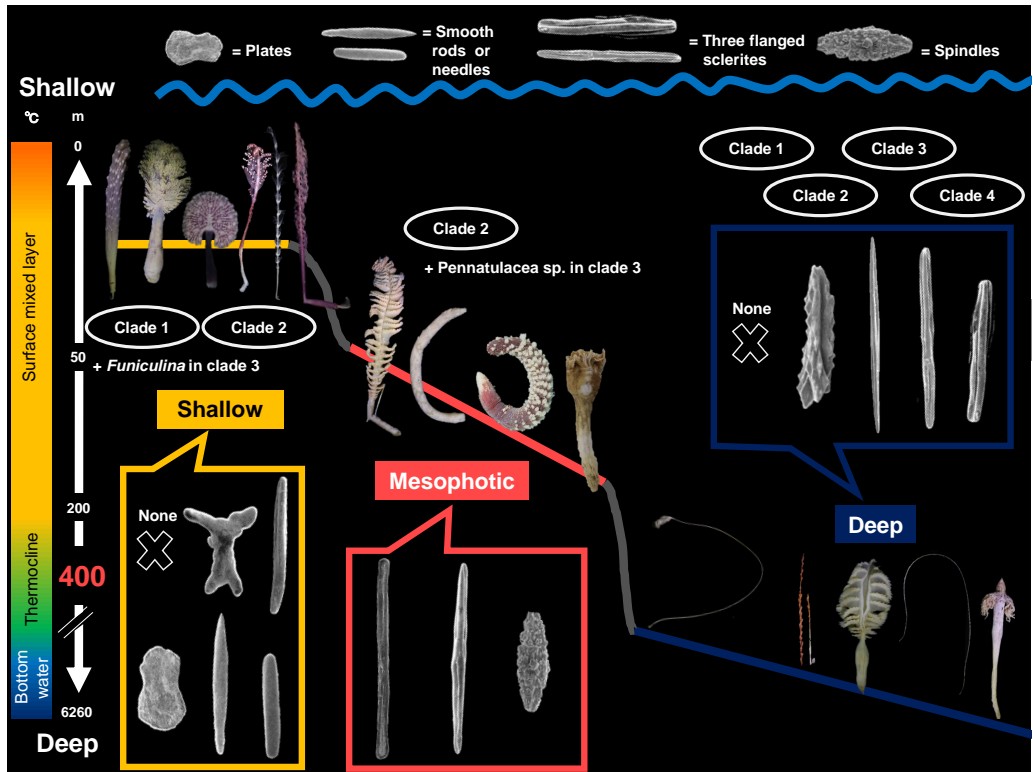

**Figure 8  Summary of this study.** This study demonstrates that the diversification in order Pennatulacea is related to habitat depths and suggests sclerite shape may be affected by the surrounding environmental factors such as water temperature. Sclerites in each box indicate the representative sclerite shapes. Bar on the left indicates metalimnions of water temperature (°C). Bar on the right indicates depth (m).

Changing ocean conditions across time likely prompted changing habitat depths in various sea pen species, and these depth shifts may be related to the evolution of sclerites in sea pens. To resolve this hypothesis, time divergence estimation is an effective method to trace the evolution of sea pens. It is known that fossil records of sea pens are sparse and almost completely consist of axes (*Reich & Kutscher, 2011*). Furthermore, the first confirmed fossil record of a sea pen is from the Cretaceous (70–83 mya; *Anderson, 1979*; *Reich & Kutscher, 2011*). Interestingly, a divergence time calculation of sea pens studied based on mtMutS, COI, and 28SrDNA sequences by *García-Cárdenas, Núñez Flores & López-González (2020)* estimated the origin of sea pens to be in the lower Cretaceous (∼144 Ma). However, this estimate is far more recent than robust estimates by *Quattrini et al. (2020)* and *McFadden et al. (2021)*, who suggested 300 Ma ∼400 Ma with time-calibrated phylogenetic trees from hundreds of ultra-conserved elements and exon loci. Combining the morphological data of sea pens with genome-wide phylogenetic analyses is a logical next step in understanding the evolution of sea pen forms.

# ACKNOWLEDGEMENTS

The authors thank all researchers and the crew of the T/RV *Seisui-maru* (Mie University) cruise 1903, all researchers and crew of the R/V *Kaimei*, and operators of the KM-ROV during the KM19-05C cruise, staff of Okinawa Churaumi Aquarium (Okinawa Churashima Foundation), Honmei Kishimoto, captain of the *Daini-Kuroshio-Maru*, and the Motobu Fisheries Cooperative, all staff of Shimoda Marine Research Center (University of Tsukuba), Amakusa Marine Biological Laboratory (Kyushu University), Dr. Hiroki Kise (National Institute of Advanced Industrial Science and Technology) and Dr. Naoto Jimi (Nagoya University) for specimen collection, all technical staff in Vigo University (Spain) for sequencing, Dr. Euichi Hirose (University of the Ryukyus) for access to SEM. We also thank Dr. Claudio Vasapollo for editing this manuscript, and the reviews of Dr. Les Watling and two anonymous reviewers that greatly improved this manuscript.

### Funding

Yuka Kushida's visit to the Western Australia Museum was supported by funding from Okinawa Research Core for Highly Innovative Discipline Science from the University of the Ryukyus. The sub-Antarctic specimens were collected by the ACE expedition supported by the ACE Foundation and Ferring Pharmaceuticals and supplied by the Western Australian Museum. This work was supported by the research project Tohoku Ecosystem-Associated Marine Sciences from the Ministry of Education, Culture, Sports, Science, and Technology (grant Number: JPMXD1111105260) and the common university budget of James Davis Reimer. The funders had no role in study design, data collection and analysis, decision to publish, or preparation of the manuscript.

### Grant Disclosures

The following grant information was disclosed by the authors:
University of the Ryukyus.
ACE Foundation and Ferring Pharmaceuticals: Western Australian Museum.
Ministry of Education, Culture, Sports, Science, and Technology: JPMXD1111105260.

### Competing Interests

James Davis Reimer is an Academic Editor for PeerJ.

### Author Contributions

- Yuka Kushida conceived and designed the experiments, performed the experiments, analyzed the data, prepared figures and/or tables, authored or reviewed drafts of the article, and approved the final draft.
- Yukimitsu Imahara performed the experiments, authored or reviewed drafts of the article, and approved the final draft.
- Hin Boo Wee performed the experiments, analyzed the data, authored or reviewed drafts of the article, and approved the final draft.

- Iria Fernandez-Silva performed the experiments, analyzed the data, authored or reviewed drafts of the article, and approved the final draft.
- Jane Fromont performed the experiments, authored or reviewed drafts of the article, and approved the final draft.
- Oliver Gomez performed the experiments, authored or reviewed drafts of the article, and approved the final draft.
- Nerida Wilson performed the experiments, authored or reviewed drafts of the article, and approved the final draft.
- Taeko Kimura performed the experiments, authored or reviewed drafts of the article, and approved the final draft.
- Shinji Tsuchida performed the experiments, authored or reviewed drafts of the article, and approved the final draft.
- Yoshihiro Fujiwara performed the experiments, authored or reviewed drafts of the article, and approved the final draft.
- Takuo Higashiji performed the experiments, authored or reviewed drafts of the article, and approved the final draft.
- Hiroaki Nakano performed the experiments, authored or reviewed drafts of the article, and approved the final draft.
- Hisanori Kohtsuka performed the experiments, authored or reviewed drafts of the article, and approved the final draft.
- Akira Iguchi performed the experiments, authored or reviewed drafts of the article, and approved the final draft.
- James Davis Reimer conceived and designed the experiments, performed the experiments, authored or reviewed drafts of the article, and approved the final draft.

**Field Study Permissions**

The following information was supplied relating to field study approvals (i.e., approving body and any reference numbers):

Field experiments which needs the permission conducted under Ministry of Agriculture, Forestry and Fisheries.

**DNA Deposition**

The following information was supplied regarding the deposition of DNA sequences:

The sequences are available at Genbank: ON586695– ON586729 and ON603941– ON603971.

**Data Availability**

The raw data is available in the Supplementary Files, tables, and figures.

**Supplemental Information**

Supplemental information for this article can be found online at http://dx.doi.org/10.7717/peerj.13929#supplemental-information.

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
