# Peer review of "Exploring the trends of adaptation and evolution of sclerites with regards to habitat depth in sea pens"

_PeerJ, doi:10.7717/peerj.13929_

## Round 0.1 · original submission · Minor Revisions

All three reviewers agree on the importance of the work made, but 2 of them (Rev 1 and 3) have some perplexities on the speculative nature of the discussions, not always supported by clear and strong evidences. Please, clearly and deeply read their comments. According to rev 2, it should be an added value to discuss the importance of the finding on the light of conservation measures on organisms facing several threat (e.g, trawling, environemental modification, and so on). Last but not least, read carefully what rev 3 commented about CCD.

Reviewers 1 & 2 have suggested that you cite specific references. You are welcome to add it/them if you believe they are relevant. However, you are not required to include these citations, and if you do not include them, this will not influence my decision.

Here my personal comments.
Line 208: Can you explain what does GTR+G+I stand for? And also HKY+G model
Line 233: "...depth groups (n=2...)" According to what you wrote at lines 178-180, I expected more depth subdivision. Moreover, if you want to keep these two groups, please can you specify which was the cutoff to distinguish shallower from deeper?
Line 245: Please, report in MM what this ratio stands for and how it should be interpreted for those that are not familiar with this approach.
Line 307: "...or 4 (n=0)." More than rare it is completely absent.
Line 310: I would reduce the decimals to 2 for the X2 as well as for std residuals and to three for p values.
Lines 397-423: According to the title of the MS I suppose this subparagraph should be reported at the beginning of the Discussion. Moreover, possibly the term evolution could be omitted here as it is more deeply evaluated in the other subparagraph taking into consideration the evolution of slcerites.
Line 440: I think this part also could be moved at the beginning just after the subparagraph Adaptation and evolution...and so on.
Table captions: add some details to the captions considering the tables themselves as standalone object. specify the depth groups deriving from, for example, as well as the clades.
Figure 5 caption: finally I understood the depth groups. So, I strongly reccomend to specify it in MM. Moreover, you should write how did you choose this grouping. Why 0-400 and 400-6100? Why no intermediate depths between 400 and 6100? Also, why didn't you use the depth categorization instead of the depth groups for your analysis since you presented them in this figure?
Figure 6: please, be consistent. In the image you wrote Ten, while in the caption you reported ten. Change according to the image. Moreover, change Dark purple in dark blue as it seems so, even if I don't understand th eimportance of specifying this color as the image considers only one family.
Figure 8: what does the X stands for in the clouds?

Reviewer 1 ·

Basic reporting

Good presentation for the most part: clearly written with professional English usage, nicely structured with good quality of figures presented.

Experimental design

Well-presented overall (also see Additional Comments below).

Validity of the findings

In my view, this is the weakest aspect of the paper. In many cases, underlying data is not robust, much conjecture and speculation is presented without sound reasoning, supporting data, evidence, or reference to previous publications (also see Additional Comments below).

Additional comments

Conclusions: Because of the frequent use of speculation and few good examples of supporting evidence, I see this manuscript as more of review paper or synopsis of various aspects of our knowledge of a restricted selection of overall sea pen diversity, with speculation offered as possible explanations for bathymetric influence on pennatulacean evolution. Perhaps something to this effect should be included in the Abstract, in order to more clearly define the intent of the paper.
Specific Comments:
Line 78, .... remains limited (delete lacking).
Line 149, Between the second to last sentence and the last sentence of this paragraph, add this sentence: Variability of sea pens and the transition to the deep-sea was treated in Pasternak, 1989.
Regarding “Evolution of sea-pen sclerites across habitat depths”, beginning on line 440. I am concerned that a significant amount of suggestion, speculation, or conjecture is put forth here without sufficient supporting evidence or citations. In addition, the authors are using a limited number of pennatulacean taxa in their arguments. Some examples: Relevant to lines 458-459 and lines 472-475, there is no hard evidence to suggest that three-flanged sclerites originated in the deep-sea or is due to physical pressure. Examples of sea pen genera with three-flanged sclerites that inhabit <400 m in depth are Actinoptilum, Echinoptilum, Renilla, Ptilosarcus, Grasshoffia, and Acanthoptilum. Reference to these are absent or not well covered in the text. Conversely, examples of genera that do not have sclerites that inhabit depths between 1000 and 6000 m include Umbellula, Porcupinella, Anthoptilum, Chunella, and Amphiacme. I believe a statement needs to be included perhaps in the abstract or introduction to the effect that much of this paper relies on conjecture and speculation with only limited number of examples of pennatulacean taxa or lack of evidence to back up the claims made.
Line 534, There are three deep water genera (family Chunellidae) that are not in your Clade 1 or treated in your manuscript. These are Amphiacme, Chunella, and Porcupinella – (see: Williams, 1990 and Williams, 2021).
Line 568: Lack of supporting evidence or cited previous studies to make this claim.
Between lines 743 and 744, add: Pasternak, F.A. 1989. On the variability of sea-pens (Octocorallia: Pennatulacea) connected with the transition to the deep-water existence. Trudy Instituta Okeanologii, Akademii Nauk SSSR 124: 165-173. [in Russian with English Summary]
Between lines 765 and 766, add: Williams GC. 1990. The Pennatulacea of southern Africa (Coelenterata, Anthozoa). Annals of the South African Museum 99(4): 31 119.
Between lines 773 and 774, add: Williams GC. 2021. The deep-sea pennatulacean genus Porcupinella – with the description of a new species from Tasmania (Anthozoa, Octocorallia, Chunellidae). Zookeys 1019: 1-14. https://doi.org/10.3897/zookdys. 1019.61789.

Reviewer 2 ·

Basic reporting

The paper "Exploring the trends of adaptation and evolution of sclerites with regards to habitat depth in sea pens" is a very interesting piece on a scarcely studied order (Pennatulacea) of octocorals.
The paer is very well structured, easy to read and stated clear research questions.
References are updated and complete but I suggest to consider also these two contributions in the introduction, when the plasticity of sclerites is reported as affected by environental conditions. Cerrano et al., 2013. Red coral extinction risk enhanced by ocean acidification. Scientific reports, 3(1), pp.1-7. and Rastelli et al., 2020. A high biodiversity mitigates the impact of ocean acidification on hard-bottom ecosystems. Scientific reports, 10(1), pp.1-13.
The paper stimulates and suggests several possible answers to the research questions, widening evolutionary interpretation of sea pens differences along a depth gradient.

Experimental design

The design is correct and the available details of the analyses allow their replicability

Validity of the findings

The discussion on presence/absence of sclerites is well organized and cover many aspects. Sea level rise in the last 12000 years has shaped the distribution of benthic species and may be some recent adaptation to shallow waters.
Line 477-48, please provide some depth range to better clarify compensation depths.

Additional comments

Just a general consideration on the urgency of these studies. Sea pens distribution in soft bottoms is a serious problem for their conservation and knowledge. Their habitats are among the most exploited by fishing trawling activity and we are loosing wide and entire population of sea fans all over the world. OSPAR convention include them among essential fish habitats and vulnerable habitats and GFCM in the Mediterranean Sea is considering their presence as one of the element to consider in the design of conservation measures. I think that, at least at the end of discussion, some consideration on this could be added.

·

Basic reporting

All these criteria are met. The paper is very well produced, excellent figures, etc.

Experimental design

no comment

Validity of the findings

See below for detailed comments.
In general, however, this is a very interesting and stimulating paper and will act as the basis for much further research.
You will see in the detailed comments below where I either disagree with statements made or have suggestions for additional comparisons, etc. In all cases, you may have counter-arguments and can rebut what I have written.
My major objections are in the Discussion. A lot is speculative, and some is incorrect in my view, but in any case the Discussion is too long and much is not needed in support of the primary thesis of the paper.

Additional comments

Fig. 6 legend doesn’t seem to reflect the figure. No purple color.
Fig. 8. Legend needs to be much more elaborate so the fig. can stand alone. One could at least indicate what the order of sclerites in the boxes indicates.
I would much prefer the authors to use temperatures rather than shallow vs. deep. Shallow waters at the poles are much colder than shallow waters at the tropics, and that may explain some of the shallow anomalies. Yes, depth is important, especially when one gets to bathyal depths and food starts to become limiting.
l. 378-395. People tend to accept the names of the taxa and then say the taxon is polyphyletic. I would rather the authors accept the idea that the specimens in question are either mis-identified or the taxa in question are inadequately defined. It would seem to me that much could be done with the cladogram in Fig. 5 to undertake a massive re-arrangement of species and genera, if the molecular evidence presented is to be believed, and I don’t see any reason why it shouldn’t be. So, that means that sea pen taxonomy is a mess and should be reconstructed based on the better genetic evidence and that all the families need to be re-diagnosed, perhaps with new characters.
Suppl. figures need legends. Otherwise who knows what is being presented.
l. 305-328 and l. 419-423. I don’t see the point of the chi-square analysis. All the clades have some deep species and shallow species are mostly in clades 1 and 2, with a small number in clade 3. It is possible that diversification into the various clades happened in the deep and that there was some emergence to shallow water in all but clade 4. One can see that in Fig. 6, so I am not sure why a statistical analysis needs to be added. Same with sclerite shape.
L. 429. Those serrated edge sclerites also resemble the sclerites found in the pharyngeal wall of Keratoisididae.
L. 432. Reconstruct this sentence. The current study is hypothesizing that….
L. 436-438. Sea pens pre-date keratoisidids, but as Heestand-Saucier et al. showed, are not so distant from keratoisidids, which are also a predominantly bathyal taxon, so the resemblance of the sclerites is not surprising.
l. 449-450. The sclerites depicted could easily be from a keratoisidid. We would call the sclerites from the tentacles “rods” and those rods are often found in keratoisidid bamboo corals. Interestingly, the rachis sclerites look much like the pharyngeal sclerites also found in those bamboo corals, especially the smaller ones in the lower row of Fig. 6.
l. 458-459. There are many factors differentiating the bathyal from the shallow waters of the world ocean. Physical pressure is only one and there is no evidence from physiology of other organisms that physical pressure is important until abyssal depths are reached. So, I think this is speculation with no basis in fact.
l. 465-466. “needed” or a genetic holdover? Probably not “needed” in the deep either, but is a remnant from the early ancestor of sea pens. We know virtually nothing about sclerites in octocoral phylogeny so all you can say is that the three-flange sclerite was present in the living representative of an early sea pen, but not “why.”
l. 469. CCD is at 4000 m in most oceans, so that is not likely a factor either. In fact, I have very large bamboo corals from Tasmania collected at 4000 m and they have a very large calcareous axis and large sclerites, of the same form as those in Fig. 6. On the other hand, the bamboo corals from 4km depth in the manganese nodule province are a few cm long at most. If you look at deposition of organic matter to depth in those two areas you will see that the area off Tasmania gets a lot more org C input that does the nodule area. So, I think CCD has nothing to do with it.
L. 474-475. Yes!
l. 476 ff. No to CCD and no to pressure, as already noted. Check out the issues of deep sea on physiology in this very nice text: Somero et al. 2017 Biochemical Adaptation: Response to Environmental Challenges from Life's Origins to the Anthropocene
l. 534 ff. Loss of sclerites has been seen in other shallow water octocorals, so is most likely a derived condition.
l. 565. Not “relies on” but is “related to”.
l. 567. Water temperature, yes; CCD, no.

---

## Round 0.2 · accepted · Accept

You made a great job with your corrections and the manuscript is now good enough to be published. I am satisfied with the responses given to reviewers. Congratulations again.

Reviewer 1 ·

Basic reporting

no comment

Experimental design

no comment

Validity of the findings

no comment

Additional comments

I have reviewed the corrected manuscript and the comments of the three reviewers. I believe the authors have made an adequate effort to address the comments. I have no further comments to make and therefore believe the corrected manuscript can now be accepted for publication.

·

Basic reporting

no comment

Experimental design

no comment

Validity of the findings

no comment

Additional comments

It looks to me that the authors took the review comments seriously and addressed them all. I have no additional comments.